# Dissecting Platelet’s Role in Viral Infection: A Double-Edged Effector of the Immune System

**DOI:** 10.3390/ijms24032009

**Published:** 2023-01-19

**Authors:** Hajar El Filaly, Meryem Mabrouk, Farah Atifi, Fadila Guessous, Khadija Akarid, Yahye Merhi, Younes Zaid

**Affiliations:** 1Health & Environment Laboratory, Ain Chock Faculty of Sciences, Hassan II University of Casablanca, Casablanca 20000, Morocco; 2Department of Biology, Faculty of Sciences, Mohammed V University, Rabat 10100, Morocco; 3Immunology and Biodiversity Laboratory, Department of Biology, Ain Chock Faculty of Sciences, Hassan II University of Casablanca, Casablanca 20000, Morocco; 4Faculty of Medicine, Mohammed VI University of Health Sciences (UM6SS), Casablanca 82403, Morocco; 5Department of Microbiology, Immunology and Cancer Biology, School of Medicine, University of Virginia, Charlottesville, VA 22903, USA; 6Research Center, Laboratory of Thrombosis and Hemostasis, Montreal Heart Institute, Faculty of Medicine, Université de Montréal, Montreal, QC H3T 1J4, Canada; 7Department of Medicine, Research Center of Abulcasis University of Health Sciences, Rabat 10000, Morocco

**Keywords:** platelets, thrombosis, viral infection, inflammatory function, immune response, TLRs

## Abstract

Platelets play a major role in the processes of primary hemostasis and pathological inflammation-induced thrombosis. In the mid-2000s, several studies expanded the role of these particular cells, placing them in the “immune continuum” and thus changing the understanding of their function in both innate and adaptive immune responses. Among the many receptors they express on their surface, platelets express Toll-Like Receptors (TLRs), key receptors in the inflammatory cell–cell reaction and in the interaction between innate and adaptive immunity. In response to an infectious stimulus, platelets will become differentially activated. Platelet activation is variable depending on whether platelets are activated by a hemostatic or pathogen stimulus. This review highlights the role that platelets play in platelet modulation count and adaptative immune response during viral infection.

## 1. Introduction

Platelets originating from megakaryocytes are anucleate cells that play a key role in vascular repair and maintenance of hemostasis, particularly in primary hemostasis [1]. Located in blood vessels, platelets have a discoid shape, a size of 3 mm by 0.5 mm, a lifespan of 10 days, and a count of 250 million of adult blood molecules per mL [2,3]. Platelets have traditionally been associated with rapid procoagulant responses mediated by G-protein-coupled receptors that promote platelet function including adhesion, activation, aggregation, eicosanoid synthesis, and granule secretion [4]. Platelet membrane integrins can interact with molecules of the injured endothelium, inducing their adhesion, activation, and aggregation in turn. Consequently, the formation of a thrombus takes place, and this clot consists of platelets aggregate bonded together by fibrinogen and ensuring closing the vascular breach [5]. In addition to their role in hemostasis, studies have shown that platelets can aggregate at the bacterial invasion site, accumulate in inflammatory areas, and target susceptible tissues to antigen-mediated inflammatory responses [6]. In fact, this platelet aggregation is a defense mechanism to aid pathogens clearance by the immune system [7]. When platelets cannot, extracellular vesicles derived from platelets can enter lymph, bone marrow, and synovial fluid. Consequently, platelet-derived extracellular vesicles (PEVs) are able to transfer a variety of contents to cells and organs inaccessible to platelets [8].

Because of their rapid presence at the injury site, and due to their speculated role in infectious diseases, platelets became well known as the first immune cells to be in contact with the pathogen during systemic infection. Indeed, infections are often associated with thrombocytopenia, which predicted increased severity, suggesting that these cells might have a great importance in coping with pathogens [9]. To do so, platelets must be able to activate other cells of the innate and adaptive immunity through (1) detecting the pathogen, (2) targeting it (and eliminate when possible), and (3) warning other cells about the presence of a pathogen as well as its type [10]. The interaction between platelets and infectious pathogens involves different receptors and intra-platelet signaling, leading to distinct responses depending on the pathogen [9].

All cellular components responsible for hemostasis and immunity are transferred to platelets by megakaryocytes, including chemokines, immune receptors, RNA molecules, and spliceosomes [11,12]. It has been found that megakaryocytes are susceptible to a variety of viruses [13]. Further, megakaryocytes express pattern recognition receptors (PRRs) and cytokine receptors, which affects megakaryocytic maturation and thrombopoietic activity [13,14]. In vitro, megakaryocytes respond to viral infections as well as viral pathogen-associated molecular patterns (PAMPs) by producing large amounts of IFNs, which in turn reduce platelet production through an autocrine interferon-α/β receptor (IFNAR) pathway [13,15]. The involvement of megakaryocytes in immune response still requires more investigation, even if megakaryocyte infection might alter the phenotype of platelet progeny during infections.

Through the expression of a wide variety of PRRs and hemostatic receptors, platelets are able to capture fragments of pathogens, whether they are bacteria, viruses, parasites, or fungi [9]. Precisely, platelets and their progenitor cells, the megakaryocytes (MK), possess direct antiviral immune activities and have shown the ability to internalize viruses. In fact, these unique cells ensure their immune role since they express a large number of receptors dedicated to viruses’ interaction [6]. In addition, and as a response to this interaction, these cells have the ability to secrete several inflammatory and/or immunomodulating molecules that can interact with other immune cells (or non-immune cells such as endothelial cells) and modulate the cellular responses of both innate and adaptive immunity [16]. Platelets can produce molecules involved in the adaptive response such as FasL, TRAIL, IL-7, and CD40L. The role of FasL and TRAIL in platelets has been poorly studied; however, FasL and TRAIL are known to be potential inducers of apoptosis of carcinogenic or infected cells [17]. These molecules production by activated platelets could therefore be critical for the antitumor and anti-infectious response [18]. The major actor in the interactions between platelets and other immune cells is the CD40L/CD40 pair, which has long been known to induce multiple inflammatory and immune responses [19]. As a result of the CD40L/CD40 interaction, mitogen and stress-activated protein kinase (MAPK/SAPK) cascades are activated, transcription factors are also activated, cytokines are secreted, B cells proliferate and differentiate into Ig-secreting plasma cells, and humoral memory is established [20].

As for IL-7, activated platelets was shown to be one of the major sources of this cytokine [21]. Consequently, considering IL-7 signaling during viral infection, remarkably increased numbers of T cells’ effector were noted, suggesting its role in immune cell expansion [22]. After platelets recognize the pathogen, they become activated, and the activated platelets, via various mechanisms, kill or sequester the pathogen by activating neutrophils and macrophages. As part of the innate immune response, platelet neutrophil interaction leads to neutrophil extracellular traps (NETs), which can enhance platelet adhesion, activation, secretion, and aggregation inducing microthrombi formation [23].

In this narrative review, our aim was to highlight the role of platelets in viral infection through depicting their interaction with multiple viruses, its consequence, and its way of affecting the viral-associated physiopathology.

## 2. Viral Receptors on the Platelets Surface

Platelets have emerged as one of the crucial players in mediating the response to infectious disease and especially to viruses. While platelets do not have nuclei, they possess all the molecular machinery to synthesize proteins from stored mRNA, suggesting they can translate proteins from RNA viruses as well [24,25]. On the surface of these tiny bits of cytoplasm, a variety of expressed receptors allow for their interaction with the virus [6]. Indeed, this interaction involves virus-specific receptors and surface glycoproteins whose original hemostatic function is hijacked by viruses, allowing for their recognition [6].

In both experimental viral infections and naturally infected human patients, platelet participation in immune response to virus has been investigated. Many of the PRRs associated with viral recognition have been found to be present and functional in platelets [26,27]. Platelets express various PRRs such as TLRs, complement, and Fc- γ receptors. As for TLRs, these functional PRRs are able to sense microbes, subsequently triggering platelet effector responses responsible for modulating the innate immune response [28]. Platelets and megakaryocytes express TLRs (TLR 1, TLR 2, TLR 3, TLR 4, TLR 6, TLR 7, TLR 8, and TLR 9) that detect and bind viral components on their surface and viral nucleic acids [29]. Once activated, TLRs recruit adaptor molecules are required for signal propagation to lead to the induction of genes that orchestrate inflammation [29]. TLR 4 on platelets acts as an inflammatory sentinel and surrounds and isolates an infection, as well as modulating proinflammatory cytokine release [30]. The induced response against single-stranded RNA viruses by platelets was noted to be a predominantly TLR 7-mediated process [26,31]. TLR 7 is located in platelets’ endolysosomes and requires internalization of virus particles and the acidic pH of endolysosomes for its own activation and signaling [26]. This TLR was also involved in enhancing platelets’ uptake of viruses, such as influenza, leading to neutrophil NETosis [32]. Furthermore, Koupenova et al. recently demonstrated that influenza virus engulfment through platelets causes the release of complement factor C3 and the subsequent activation of neutrophils and NETosis [31].

Similarly, Cytomegalovirus (CMV) was shown to binds to platelets through TLR2, which triggers platelet activation and secretion and results in enhanced platelet interaction with neutrophils [32,33]. On both platelet surface and in intracellular compartments, TLR3 was found to be responsible for recognizing double-stranded RNA viruses [28]. EMCV has been shown to interact with platelet TLR7, which leads to degranulation of platelets and direct interactions between platelets and neutrophils [26]. In the same manner, activated platelets express TLR9 on their surface and ensure the sequestering of viral DNA [34]. During viral infections, PARs on platelets, endothelial cells, and leukocytes modulate innate immune responses and affect TLR-dependent responses both positively and negatively [35]. The presence of other classes of PRRs involved in the viral recognition, such as retinoic acid-inducible gene I (RIG-I), was confirmed at the level of megakaryocytes when responding to type 1 interferons. However, RIG-I expression in platelets is yet to be known [28].

Platelets also express several complement receptors, such as the complement receptor type II (CR2) and Epstein–Barr virus receptor, which act as receptors for viruses that result in multiple antimicrobial defense functions, including lysis, opsonization, and chemotaxis [36]. These receptors allow platelets to capture different types of viruses. For example, GPIIb/IIIa or α2β3 integrin recognizes the RGD sequence of Adenovirus and Hantavirus. The Dendritic Cell-Specific ICAM3-Grabbing Non-Integrin (DC-SIGN) receptor contained in granules is able to bind dengue virus (DENV) when expressed on the platelet surface. Integrin α2β1 and glycoprotein GPVI (major collagen receptor) are capable of binding rotavirus VP4 protein and hepatitis C virus (HCV), respectively [6]. Platelets also express a receptor for Coxsackie viruses, the Coxsackie-Adeno Receptor (CAR) [37]. These overall receptor–virus interactions cited above are shown in Table 1.

There are two main families of platelet cytosolic sensors: NLRs, including oligomerization domain-containing nucleotide-binding domain 2 (NOD2) and leucine-rich repeat-containing pyrin 3 (NLRP3) [38,39]. A major function of the NLRP3 receptor is to activate caspase-1, which converts pro-IL-1β and pro-IL-18 into active cytokines [40]. The cytokine processing and assembly of the inflammasomes in nucleated cells are triggered by two signals: transcription of cytokines and activation of the inflammasome components [41]. A recent study has shown that platelets are activated during Chikungunya virus infection and that this can lead to the formation of NLRP3 inflammasomes and the release of inflammatory eicosanoids, cytokines, and chemokines [42].

It would also be relevant to point out that inflammation can be induced by PEVs in part due to their influence on cell–cell interactions and their involvement in inducing adhesion molecules in different types of cells and their ability to release cytokines. Additionally, PEVs contain proinflammatory cytokines like interleukin (IL)-1, IL-6, and tumor necrosis factor [43]. In COVID-19 patients, PEV-associated tissue factor activity was associated with thromboembolic events at a higher level [44,45,46]. In addition, there has also been a significant growth in circulating platelet-derived EVs, which are the major source of CD142 in plasma [47,48]. In studies on HIV and PEVs, it has been demonstrated that vesicles can facilitate viral reproduction, modify receptor expression to make cells more receptive to infection, promote viral replication and stability via host molecules, and activate latent viruses by uninfected cell EVs [49,50,51,52,53].

## 3. Role of Platelets in Antiviral Defense

Platelets interact with viruses, and their ability to internalize these pathogens started to flourish subsequently to Danon et al. discovery in 1959, where influenza virus’ internalization by platelets was observed through electron microscopy [54]. Interestingly, platelets were capable of actively extracting RNA from endothelial cells and internalizing circulating vesicles, debris, mitochondria, and pollen particles [55]. As for the newly rampant virus, SARS-CoV-2, several hypotheses regarding its internalization by platelets have been put forward [56]. Indeed, Koupenova et al. showed that these cells internalize SARS-CoV-2 following their co-incubation. As a matter of fact, three modes of entry were proposed: via endosomes, phagocytosis vacuoles, or by attachment to microparticles [57]. Other researchers suggested that the internalization of this virus by platelets is dependent on the angiotensin-converting enzyme 2 (ACE2), since both SARS-CoV and SARS-CoV-2 coronaviruses use the ACE2 receptor to infect cells [58,59]. As a second possible mechanism for viral invasion, it has been suggested that SARS-CoV-2 enters cells through the use of transmembrane serine protease-2 (TMPRSS2), which is essential for the cleavage of the SARS-CoV-2 S protein, enabling the virus to fuse with the cell membrane and be internalized [58]. Otherwise, the presence of the ACE2 receptor in platelets is debated [60]. Manne BK et al. did not detect ACE2 mRNA or protein in platelets [61]. Similarly, ACE2 receptor mRNA was not detected in platelet sequencing [45,62,63,64]. In contrast, Zhang’s team published results in favor of ACE2 expression in human platelets, where they detected both ACE2 RNA and protein in these cells [65]. Furthermore, reduced platelet count (or thrombocytopenia) is linked to increased morbidity and mortality in pandemic CoV infections [66]. Influenza virus, like SARS-CoV-2, is a single-stranded RNA virus that can infect epithelial cells, and platelets have been shown to actively internalize influenza virus particles [26,31]. It is also common for severe cases of influenza infection to present with tissue pathology and excessive inflammation and coagulation activation within the lungs [67,68]. An association between platelet accumulation in the lungs and disease progression has been demonstrated in the murine model of pulmonary viral infection [69].

In the same context, the ability of platelets to internalize human immunodeficiency virus (HIV) particles was investigated, and was first described by Zucker-Franklin et al. [70]. Incubation of platelets with lymphocyte supernatant showed that HIV was indeed internalized by these bits of cytoplasm [6]. In addition, MKs express CD4 as well as HIV co-repressors (CXCR 1, CXCR 2, CCR 3, CXCR 4, and CCR 5) [71]. On the other hand, platelets express only certain co-repressors (CXCR4 strongly, CCR1 and CCR3). Even so, and according to Youssefian et al., platelets play a dynamic role against HIV [72,73]. Nevertheless, this internalization process specificity was revisited due to White’s team who showed that platelet internalization was not dependent on the pathogen, but on the ability of platelets to spontaneously internalize particles [74]. The same study showed that the internalized particles are then in direct contact with the open canalicular system (OCS). Knowing that the granular content is discharged into the OCS during platelet activation, it is possible that this internalization, when linked to activation, could be a mechanism for platelets to make their microbicidal molecules act directly on the target [74]. Indeed, direct interaction of viruses with platelets can induce major transcriptome alteration, which activates and induces antiviral function of platelets [45,75]. The main mechanisms of interaction of platelets with viruses and the surface receptors involved in viral recognition by platelets are illustrated in Figure 1 (See legend in Appendix A).

Interestingly, ex vivo and in vitro studies showed that platelet interactions with DENV upregulate the levels of cleaved IL-1β. This upregulation was dependent on the activation and assembly of NLRP3 inflammasome components, as well as caspase-1 activation [76,77]. In addition to its involvement in the restriction of viral replication, release of IL-1β by human platelets has been linked to detrimental systemic inflammation and increased vascular permeability, resulting in shock and, eventually, death [77,78,79]. Recently, and in association with the SARS-CoV-2 pandemic, Zaid et al. revealed that platelet can associate with SARS-CoV-2 and exhibit an hyper-responsive state and express a variety of proinflammatory mediators, including ones closely related to viral responses [45,80]. In fact, the excessive production of these mediators was shown to be associated to COVID-19 severe clinical outcomes [81,82,83]. Among these proinflammatory mediators, IFN α and γ were also expressed by platelets. Beyond doubt, the interferon family is well known by its proficient antiviral role. In both severe and non-severe stages of COVID-19, platelets exhibited downregulated levels of IFN α and γ, but their overall levels in blood were still upregulated. This phenomenon might be explained by the fact that, after being internalized by platelets, SARS-CoV-2 components might cop with platelets activation [45]. Along the same lines, Manne et al. showed that the human platelet transcriptome is altered during SARS-CoV-2 infection, which induces robust gene expression and functional changes in platelets [61].

Dengue is a mosquito-borne virus that is caused by four serotypes of DENV (DENV-1 to -4). Platelets purified from patients infected with DENV and H1N1 also showed higher IFITM 3 (IFN-sensitive viral restriction factor), while its decrease was a predictor of poor survival rate [75]. Likewise, this virus was shown to induce lipid mediator synthesis and release of microparticles from platelets. This activation was mediated through the stimulation of low-affinity type 2 receptor for Fc portion of IgG (FcgRIIA) by immunoglobulin G (IgG) /H1N1 immune complexes. Interestingly, serum soluble factors, such as complement components, were also involved in activating platelets by H1N1 virus [84]. Activated platelets were shown to also release RANTES (or CCL5), a well-known chemokine, for its anti-HIV role through attracting effector cells to the lesion site [6]. In HIV/AIDS patients with high viral loads, platelet exhaustion of chemokines has been reported [85]. In addition, adenovirus contact with platelets induced their activation and rapid increase in CD62P (P-selectin) expression [86]. In response to this upregulation at the platelet membrane level, and due to the presence of its ligand P-selectin glycoprotein ligand-1 (PSGL-1) on leukocytes (monocytes and neutrophils), platelet–leukocyte aggregates were shown to form, through in vivo experiments, and resulted in an increased release of both cells microparticles [86].

Platelets do not only contribute to innate immunity, but also activate the adaptative immune system, thereby bolstering the immune response against viruses. Upon their direct interaction with HCV via the collagen receptor GPVI, platelets are activated and release the chemokine CCL5 from their α-granules. In return, positive regulation of type 1 (Th1) helper T cells against HCV infection takes place [87]. Furthermore, cytotoxic CD8+ T lymphocytes were also activated through platelets: this result was first shaped based on observations from a mouse model infected with the Hepatitis B virus (HBV) [88]. In the liver, the cytotoxic lymphocytes recruitment as well as their damage-associated activity were significantly reduced in platelets-depleted mice. Intriguingly, platelet-depleted mice who were subject to a platelet transfusion exhibited a restored lymphocyte deposition in the liver. Consequently, the severity of the viral infection-associated pathology at the level of this organ was also reestablished [88]. Otherwise, platelet gene expression was intriguingly affected by vaccination, allowing them to interface with both innate and adaptive immunity. Beyond doubt, a deep understanding of vaccination potential in modulating platelets immune activity might lead towards innovative strategies of host immunization [75].

As it has been reported, platelets were first known by their hemostatic function. Upon a viral infection, studies have focused on whether this function contributes or not to antiviral immunity. However, the duality of the hemostatic and inflammatory functions of platelets was particularly emphasized during viral infections [89]. In HBV infection, anticoagulant administration had no effect on lymphocyte recruitment. This observation mainly suggests that, independently of their hemostatic activation, platelets contribute to pathogenesis through activation of their inflammatory function [88]. In contrast, a coagulation cascade was described as one of the innate immune system components, due to its ability to diminish pathogen dissemination. During influenza infection, H1N1 was shown to activate platelets through thrombin formation [84]. Similarly, SARS-CoV-2-infected cells were showed to release extracellular vesicles (EVs) with associated tissue factor (TF) activity. The latter induces the coagulation pathway in plasma through the conversion of thrombin, which in return activates platelets by protease-activated receptor (PAR)-1 and -4 [46]. Moreover, and due to the viral microenvironment, changes occur at the endothelial level that consequently induce platelets’ adhesion and activation [89]. For example, DENV was observed to activate the endothelium through E-selectin expression, allowing platelets to bind via CD62P [89]. Endothelial cells were also activated by adenovirus infection. In return, these cells exhibited a massive release of von Willebrand factor (vWF) within two hours post-infection as well as vascular cell adhesion molecule-1 (VCAM-1), which is an another platelet adhesion protein [86]. Consequently, after the vascular accumulation of platelets, thrombus formation takes place, and this process could explain the atherothrombotic events associated with certain viral infections [6]. Other than that, the thrombin generation following platelet activation is known to amplify the inflammation [89,90], suggesting that this duality of platelets’ functions might act in a synergic way to clear the infection. Indeed, platelet reactivity to thrombin has recently been shown to differ between patients with COVID-19 and those with ARDS unrelated to COVID-19 [80]. In another recent study, it was demonstrated that the SARS-CoV-2 spike protein engaged the CD42b (or GP Ibα) receptor—the high-affinity receptor for thrombin—in two distinct ways to activate platelets and promoted platelet–monocyte communication through the engagement of P-selectin/PGSL-1 and CD40L/CD40 axes, which resulted in monocytes producing proinflammatory cytokines [91].

These overall observations suggest that platelets can contribute to limiting viral infections through both their hemostatic and inflammatory functions. As they bolster the antiviral immunity, these functions can also exacerbate the viral-associated immunopathology through the exhibited hyper-responsiveness. Platelet–virus interactions are still largely unknown, but both hosts and viruses may benefit from them. For example, when HIV interacts with platelets, they release CCL5, which recruits highly susceptible target cells like T-lymphocytes and monocytes [92]. In addition, influenza virus has been demonstrated to act as a carrier in the circulation via platelets [93,94]. The HCV infection is also taken benefit of by platelets to reach the liver, where platelet activation further enhances the interaction between the platelets and the liver cells. As a result, the time for the virus to potentially infect liver tissue is prolonged [95]. On the other hand, Nielsen et al. showed impaired platelet aggregation and rebalanced hemostasis in patients with chronic HCV infection [96].

Viruses, in turn, batten down their hatches to evade the platelet-induced responses through various mechanisms. For instance, the platelet–virus interaction remains to be classified as beneficial or detrimental. After being actively internalized by platelets, HCV seems to be protected from the host immune system. Therefore, this virus uses platelets as a safe transport system to reach the liver, where it infects and damages the hepatic tissue, allowing for its sustained dissemination throughout the body [6,95]. Similarly, DENV was shown to use platelet translational machinery to replicate its viral RNA and thereby taking advantage of this immune frontline [79]. Its internalization by CD61-expressing platelets was suggested as a viral protection mechanism. In response to this internalization, monocytes phagocytose these infected platelets. Intriguingly, these phagocytes were described for their inability to neutralize the virus as it is protected within the platelet [28]. Viruses can also cope with platelet-induced immune response through thrombocytopenia, which is a hematologic disorder frequently detected in several viral infections. Several mechanisms have been proposed to contribute to thrombocytopenia in HCV infection: Sequestration of platelets within enlarged spleen [97] via platelet-associated IgG-mediated destruction, which leads towards platelet sequestration in the reticuloendothelial system as well as to hypersplenism [97,98,99]. In addition, alteration of hepatic thrombopoietin production [100,101,102] and a direct viral effect are involved, since a positive correlation between thrombocytopenia and the association of HCV with platelets has been found [103]. Interestingly, an obvious similarity between one of HCV peptides and amino acids 49–66 of the platelet surface integrin GP IIIa has been identified, supporting the idea that HCV might use molecular mimicry to its favor, leading to generation of platelet autoantibodies [104]. High-affinity binding of HCV to the platelet membrane, followed by anti-HCV antibody binding, could also promote the formation of circulating “anti-HCV antibody–antigen-platelet” complexes, leading to the phagocytosis of platelets as “innocent bystanders” [105]. Similarly to HCV, αIIbβ3 (or GP IIbIIIa) revealed a structural similarity with retrovirus GP120 antigen, resulting in a cross reaction. In return, immune thrombocytopenic purpura develops due to a reduced platelet count [28]. Moreover, DENV nonstructural (NS) proteins released from platelets in the plasma induce antiplatelet autoantibodies and, thereby, thrombocytopenia. In fact, the production of viral antigens is believed to impact platelet clearance through immune complex formation [79]. DENV can also act on platelet clearance through inducing their apoptosis [79]. DENV-infected platelets support an abortive viral infection, in which the viral genome is translated and replicated but no replicas are released [106]. Patients with dengue have been shown to have increased platelet activation associated with a more severe disease state [107,108]. When dengue infection occurs, platelet count declines correlate with expression of surface markers of activation, including P-selectin and CD63, and exposure to phosphatidylserine [39,109]. An increase in platelet activation may contribute to platelet loss by deposition of platelets in the peripheral microvascular bed [39]. In vitro, DENV-infected endothelial cells exhibited platelet adhesion to platelets, which led to a reciprocal increase in platelet activation [89]. In dengue patients, platelet–leukocyte aggregates have also been found on monocytes, lymphocytes, and granulocytes, suggesting that activated platelets and leukocytes may play a role in thrombocytopenia [77]. Dengue patients have higher levels of platelet–monocyte aggregates in their circulation, which negatively correlates with their platelet counts [77]. In closing, viruses can suppress immunity through establishing latency, which is shown to be insured by viral miRNAs. For example, human cytomegalovirus (HCMV) miRNA at the plasmatic level, hcmv-miR-US25-2-3, was negatively associated with pro-thrombotic P-selectin changes, which is expressed on the activated platelets, suggesting that dysregulation of the host’s hemostatic function may be required for efficient viral latency through immune suppression [55].

## 4. Conclusions

Platelets are essential for vascular repair and maintenance of hemostasis, but they also play an important role in immunity by expressing numerous integrins as well as cytokine/chemokine receptors. Platelets are increasingly recognized as immune cells due to new platelet functions emerging over time. Platelets are now known to interact with all types of pathogens and most importantly viruses. Indeed, the platelet response, thought to be only simple but effective in hemostasis, is for sure extremely complex and targeted in inflammatory and immune responses. In order to gain a clear understanding of antiplatelet therapies’ effects on viral infections, further studies are needed to explain the role of platelets in viral infections. By studying platelets during viral infections, we will be able to predict whether they will be beneficial or detrimental.

## Figures and Tables

**Figure 1 ijms-24-02009-f001:**
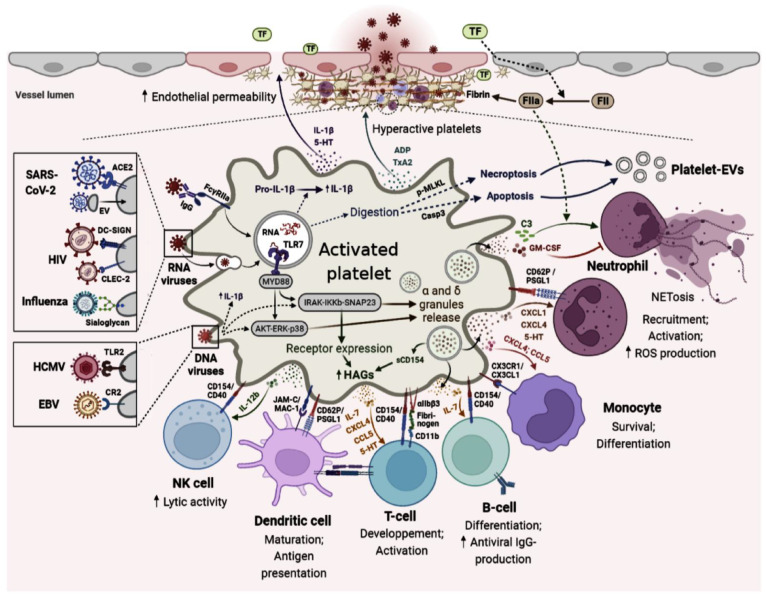
Graphical illustrating platelet immune mechanisms during viral infection.

**Table 1 ijms-24-02009-t001:** Overview of platelet receptors involved in platelet–virus interactions according to Flaujac et al. [6].

Platelet Receptors	GPVI	DC-SIGN	CCR1, CCR3, CCR4, CXCR4	CR2	α2β1	CLEC-2	α2β3 (RGD)	CAR
Virus	HCV	Lentivirus HIV	HIV	EBV	Rotavirus (VP4)	HIV-1	Adenovirus penton base Hantavirus	Adenovirus

## Data Availability

Not applicable.

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
