# Peer review of "Dissecting Platelet’s Role in Viral Infection: A Double-Edged Effector of the Immune System"

_ijms, 2023, doi:10.3390/ijms24032009_

Round 1

Reviewer 1 Report

Please, correct Abstract sentences and Keywords, excuse me.

Author Response

Please, correct Abstract sentences and Keywords

We thank the reviewer for these relevant corrections.

Reviewer's suggestions in abstract and keywords have been added to the manuscript (highlighted in yellow)

Reviewer 2 Report

Platelet is known as a hemostatic effector. Actually, platelet has immune function. This review described the role of platelet during viral infections. It could interact with many immune cells againt virus, while it could be used by virus as invading vehicle. Platelet could also be activated during infections.

When virus invaded host, many molecules and signals took part in the immune process related with platelet, which was quite complex. The review tried to explain the process from viral receptors to defending process.  But the defending process was not narrated based on the viral receptors, or on the signal pathway.

The essay structure needs reconsideration. The figure summarized the platelet’s immune mechanism, including viral receptors and kinds of immune cells. The figure also showed the interactions between platelet and virus, as well as interactions between platelet with immune cells. The related contents were mixed in the first part. It might be better to be discussed separately. Besides, the activation of platelet sometimes happened during viral infection, resulting in thrombus of different severity. It was also described in the review, which worth an in-dependant chapter.

Author Response

The reviewer 2

Platelet is known as a hemostatic effector. Actually, platelet has immune function. This review described the role of platelet during viral infections. It could interact with many immune cells against virus, while it could be used by virus as invading vehicle. Platelet could also be activated during infections.

When virus invaded host, many molecules and signals took part in the immune process related with platelet, which was quite complex. The review tried to explain the process from viral receptors to defending process.  But the defending process was not narrated based on the viral receptors, or on the signal pathway.

The essay structure needs reconsideration. The figure summarized the platelet’s immune mechanism, including viral receptors and kinds of immune cells. The figure also showed the interactions between platelet and virus, as well as interactions between platelet with immune cells. The related contents were mixed in the first part. It might be better to be discussed separately. Besides, the activation of platelet sometimes happened during viral infection, resulting in thrombus of different severity. It was also described in the review, which worth an independent chapter.

Response:

We thank the reviewer for taking the time to give constructive feedback on our manuscript. Regarding this reviewer’s comment, we will address all the comments point-by-point.

1-When virus invaded host, many molecules and signals took part in the immune process related with platelet, which was quite complex. The review tried to explain the process from viral receptors to defending process.  But the defending process was not narrated based on the viral receptors, or on the signal pathway.

Response

Besides their role in the processes of primary hemostasis and pathological thrombosis, platelets have emerged as key players in the immune defense. Indeed, in the mid-2000s, several studies expanded the critical role of these small blood anucleate cells, placing them in the "immune continuum" and thus changing the understanding of their function in both innate and adaptive immune responses.

Our review narrated this platelets’ complex role by highlighting the interaction between platelets and viruses on one hand; and between platelets and immune cells on the other hand, thus placing these cells at the junction of the axis virus-platelets-immune response during infectious diseases.

Therefore, we tackled the goal of this review from 2 perspectives: Firstly, we highlighted the role of platelets through dissecting their interaction with multiple viruses. Secondly, we summarized the role of platelets in anti-viral response, using a few examples of viruses, as summarized in Figure 1.

2-The review tried to explain the process from viral receptors to defending process.  

To remain in line with the aim of our review, we put the emphasis on platelets’ activation through their viral receptors such as TLRs and NLRs, among others.

Through the expression of a wide variety of PRRs and hemostatic receptors, platelets are able to capture fragments of pathogens (DAMPs), whether they are bacteria, viruses, parasites or fungi 1.

Upon such interactions, platelets can produce molecules involved in the adaptive response such as FasL, TRAIL, IL -7 and CD40L. While the role of FasL and TRAIL in platelets has been poorly studied; they are known to be potential inducers of apoptosis of carcinogenic or infected cells 2.

On the other hand, The CD40L/CD40 interaction activates mitogen-  and stress-activated protein kinase (MAPK/SAPK) pathway leading to transcription factors activation and their translocation to the nucleus followed by cytokines secretion (e.g. IL-7), B cell proliferation and differentiation into Ig-secreting plasma cells.

Platelets also express various PRRs such as TLRs (TLR 1, TLR 2, TLR 3, TLR 4, TLR 6, TLR 7, TLR 8, and TLR 9) that sense and bind viral components on their surface and viral nucleic acids 3. Once activated, TLRs recruit adaptor molecules required for signal propagation to lead to the induction of genes that orchestrate inflammation response 3, either via MyD88-dependent pathway or MyD88-independent pathway (see Figure 1 attached).

Moreover, there are two main families of platelet cytosolic sensors : NLRs, including oligomerization domain-containing nucleotide-binding domain 2 (NOD2) and leucine-rich repeat-containing pyrin 3 (NLRP3) 4 5. A major function of the NLRP3 receptor is to activate caspase-1, which converts pro-IL-1β and pro-IL-18 into active cytokines, through inflammasome components activation 6 7.

Above, we listed a few examples of platelet activated receptors by virus epitopes, leading to MAPK/SAPK, TLRs or NLRs downstream signaling pathways that are well-described in the literature.

3-The essay structure needs reconsideration. The figure summarized the platelet’s immune mechanism, including viral receptors and kinds of immune cells. The figure also showed the interactions between platelet and virus, as well as interactions between platelet with immune cells. The related contents were mixed in the first part. It might be better to be discussed separately.

Recent literature has highlighted the pivotal role of platelets in the immune response beyond their well-established role in primary hemostasis and pathological thrombosis processes. Therefore, we tried our best to summarize in our figure this complex cross-talk between common viruses (SARS-CoV-2, HIV, Influenza virus, HCMV, EBV), platelets and immune cells. Platelets activation by a given virus, triggered the release of granules content and the expression of several cytokines, chemokines, adhesion molecules, lipid mediators, co-stimulatory signals, coagulation cascade molecules, PEV,… thus contributing to an anti-viral immunity carried out by recruited immune cells to the lesion site.

Besides, the activation of platelet sometimes happened during viral infection, resulting in thrombus of different severity. It was also described in the review, which worth an independent chapter.

We totally agree with this comment regarding such process; and to remain more concise and avoid any redundancy with our previously published papers on this topic 8-10, we already included in this manuscript a few references related to this matter, as listed in the paragraph below.

Recently, and in association with the SARS-CoV-2 pandemic, Zaid et al. revealed that platelet can associate with SARS-CoV-2 and exhibit an hyper-responsive state and express a variety of pro-inflammatory mediators, including ones closely related to viral responses and coagulopathy 9 10.

In HIV/AIDS patients with high viral loads, platelet exhaustion of chemokines has been reported 11. In addition, adenovirus contact with platelets induced their activation and rapid increase in CD62P (P-selectin) expression 12. In response to this upregulation at the platelet membrane level, and due to the presence of its ligand P-selectin glycoprotein ligand-1 (PSGL-1) on leukocytes (monocytes and neutrophils), platelet-leukocyte aggregates were shown to form, and resulted in an increased release of both cells microparticles 12.

During HBV infection, coagulation cascade was described as one of the innate immune system components, due to its ability to diminish pathogen dissemination 13.

During influenza infection, H1N1 was shown to activate platelets through thrombin formation 14.

Similarly, SARS-CoV-2-infected cells were showed to release extracellular vesicles (EVs) with associated Tissue Factor (TF) activity. This latter induces coagulation pathway in plasma through conversion of thrombin, which in return activates platelets by protease-activated receptor (PAR) -1 and -4 8. Moreover, and due to the viral microenvironment, changes occur at the endothelial level, which consequently induce platelets adhesion and activation 15.

For example, DENV was observed to activate the endothelium through E-selectin expression, allowing platelets to bind via CD62P 15. Similarly, Dengue patients have higher levels of platelet–monocyte aggregates in their circulation, which negatively correlates with their platelet counts 16.

Endothelial cells were also activated by adenovirus infection. In return, these cells exhibited a massive release of von Willebrand factor (vWF) within two hours post-infection as well as vascular cell adhesion molecule-1 (VCAM-1), which is an another platelet adhesion protein 12.

Human cytomegalovirus (HCMV) miRNA at the plasmatic level (hcmv-miR-US25-2-3) was negatively associated with pro-thrombotic P-selectin changes, which is expressed on the activated platelets 17.

References

  1. Clemetson KJ. Platelets and pathogens. Cell Mol Life Sci 2010;67(4):495-8. doi: 10.1007/s00018-009-0204-2 [published Online First: 2009/11/26]
  2. Yang L, Wang L, Zhao CH, et al. Contributions of TRAIL-mediated megakaryocyte apoptosis to impaired megakaryocyte and platelet production in immune thrombocytopenia. Blood 2010;116(20):4307-16. doi: 10.1182/blood-2010-02-267435 [published Online First: 2010/07/31]
  3. Cognasse F, Nguyen KA, Damien P, et al. The Inflammatory Role of Platelets via Their TLRs and Siglec Receptors. Front Immunol 2015;6:83. doi: 10.3389/fimmu.2015.00083 [published Online First: 2015/03/19]
  4. Zhang S, Zhang S, Hu L, et al. Nucleotide-binding oligomerization domain 2 receptor is expressed in platelets and enhances platelet activation and thrombosis. Circulation 2015;131(13):1160-70. doi: 10.1161/CIRCULATIONAHA.114.013743 [published Online First: 2015/04/01]
  5. Hottz ED, Bozza FA, Bozza PT. Platelets in Immune Response to Virus and Immunopathology of Viral Infections. Front Med (Lausanne) 2018;5:121. doi: 10.3389/fmed.2018.00121 [published Online First: 2018/05/16]
  6. Shi J, Zhao Y, Wang K, et al. Cleavage of GSDMD by inflammatory caspases determines pyroptotic cell death. Nature 2015;526(7575):660-5. doi: 10.1038/nature15514 [published Online First: 2015/09/17]
  7. Hottz ED, Lopes JF, Freitas C, et al. Platelets mediate increased endothelium permeability in dengue through NLRP3-inflammasome activation. Blood 2013;122(20):3405-14. doi: 10.1182/blood-2013-05-504449 [published Online First: 2013/09/07]
  8. Puhm F, Allaeys I, Lacasse E, et al. Platelet activation by SARS-CoV-2 implicates the release of active tissue factor by infected cells. Blood Adv 2022;6(12):3593-605. doi: 10.1182/bloodadvances.2022007444 [published Online First: 2022/04/21]
  9. Zaid Y, Puhm F, Allaeys I, et al. Platelets Can Associate with SARS-Cov-2 RNA and Are Hyperactivated in COVID-19. Circ Res 2020 doi: 10.1161/CIRCRESAHA.120.317703 [published Online First: 2020/09/18]
  10. Zaid Y, Guessous F, Puhm F, et al. Platelet reactivity to thrombin differs between patients with COVID-19 and those with ARDS unrelated to COVID-19. Blood Adv 2021;5(3):635-39. doi: 10.1182/bloodadvances.2020003513 [published Online First: 2021/02/10]
  11. Holme PA, Muller F, Solum NO, et al. Enhanced activation of platelets with abnormal release of RANTES in human immunodeficiency virus type 1 infection. FASEB J 1998;12(1):79-89. doi: 10.1096/fasebj.12.1.79 [published Online First: 1998/01/23]
  12. Othman M, Labelle A, Mazzetti I, et al. Adenovirus-induced thrombocytopenia: the role of von Willebrand factor and P-selectin in mediating accelerated platelet clearance. Blood 2007;109(7):2832-9. doi: 10.1182/blood-2006-06-032524 [published Online First: 2006/12/07]
  13. Iannacone M, Sitia G, Isogawa M, et al. Platelets prevent IFN-alpha/beta-induced lethal hemorrhage promoting CTL-dependent clearance of lymphocytic choriomeningitis virus. Proc Natl Acad Sci U S A 2008;105(2):629-34. doi: 10.1073/pnas.0711200105 [published Online First: 2008/01/11]
  14. Boilard E, Pare G, Rousseau M, et al. Influenza virus H1N1 activates platelets through FcgammaRIIA signaling and thrombin generation. Blood 2014;123(18):2854-63. doi: 10.1182/blood-2013-07-515536 [published Online First: 2014/03/26]
  15. Krishnamurti C, Peat RA, Cutting MA, et al. Platelet adhesion to dengue-2 virus-infected endothelial cells. Am J Trop Med Hyg 2002;66(4):435-41. doi: 10.4269/ajtmh.2002.66.435 [published Online First: 2002/08/08]
  16. Hottz ED, Medeiros-de-Moraes IM, Vieira-de-Abreu A, et al. Platelet activation and apoptosis modulate monocyte inflammatory responses in dengue. J Immunol 2014;193(4):1864-72. doi: 10.4049/jimmunol.1400091 [published Online First: 2014/07/13]
  17. Koupenova M, Clancy L, Corkrey HA, et al. Circulating Platelets as Mediators of Immunity, Inflammation, and Thrombosis. Circ Res 2018;122(2):337-51. doi: 10.1161/CIRCRESAHA.117.310795 [published Online First: 2018/01/20]

Round 2

Reviewer 2 Report

N/A